# Formation of Potential Heterotic Groups of Oat Using Variation at Microsatellite Loci

**DOI:** 10.3390/plants10112462

**Published:** 2021-11-15

**Authors:** Michaela Havrlentová, Katarína Ondreičková, Peter Hozlár, Veronika Gregusová, Daniel Mihálik, Ján Kraic

**Affiliations:** 1National Agricultural and Food Center, Research Institute of Plant Production, Bratislavská Cesta 122, 92168 Piešťany, Slovakia; katarina.ondreickova@nppc.sk (K.O.); peter.hozlar@nppc.sk (P.H.); daniel.mihalik@nppc.sk (D.M.); jan.kraic@nppc.sk (J.K.); 2Department of Biotechnology, Faculty of Natural Sciences, University of Ss. Cyril and Methodius, Námestie Jozefa Herdu 2, 91701 Trnava, Slovakia; gregusova4@ucm.sk

**Keywords:** oat, seed parameter, microsatellite polymorphism, heterotic group

## Abstract

An evaluation of polymorphism at the microsatellite loci was applied in distinguishing 85 oat (*Avena sativa* L.) genotypes selected from the collection of genetic resources. The set of genotypes included oats with white, yellow, and brown seeds as well as a subgroup of naked oat (*Avena sativa* var. *nuda* Koern). Variation at these loci was used to form potential heterotic groups potentially used in the oat breeding program. Seven from 20 analyzed microsatellite loci revealed polymorphism. Altogether, 35 microsatellite alleles were detected (2–10 per locus). Polymorphic patterns completely differentiated all genotypes within the subgroups of white, brown, and naked oats, respectively. Only within the greatest subgroup of yellow genotypes, four pairs of genotypes remained unseparated. Genetic differentiation between the oat subgroups allowed the formation of seven potential heterotic groups using the STRUCTURE analysis. The overall value of the fixation index (F*_st_*) suggested a high genetic differentiation between the subgroups and validated a heterotic grouping. This approach can be implemented as a simple predictor of heterosis in parental crosses prior to extensive field testing or development and implementation of more accurate genomic selection.

## 1. Introduction

Cultivated oat (*Avena sativa* L.) is an economically important crop, ranking sixth in world cereal production after wheat, rice, maize, barley, and sorghum. Oat is used as green fodder, straw, hay, or silage with good balanced feed components for livestock. Oat grains are also part of humans’ diet, either directly as raw food (flakes, milk) or as raw material and ingredients in food production. They possess unique and important nutritive characteristics, particularly high contents of lipids, proteins, and micronutrients [1]. Compared with other cereals, oat grains are rich in antioxidants (e.g., α-tocotrienol, α-tocopherol, and avenanthramides) total dietary fiber, and the water-soluble dietary fiber, the β-D-glucans [2,3,4]. Generally, oat grains have multifunctional use in human nutrition, animal feeding, and in the production of health care and cosmetic products [5].

Plant breeders frequently use genetically similar parents in crosses to generate progenies to create new cultivars. Such an approach is also common among oat breeders. Elite but genetically very similar parents have the potential to generate advanced offspring without an undesirably changed genetic background. However, if breeders need to substantially extend genetic diversity and introduce new genes, they must select and cross parents genetically as different as possible. Genetically divergent parents can generate advanced hybrids with higher heterozygosity and manifested heterotic effects. Genetic diversity as well as unrelatedness is usually determined by pedigree analysis, morphological, physiological, agronomical traits, and biometric analysis. These approaches are limited mainly by low variation in these parameters and have been enriched gradually after the introduction of molecular, especially DNA markers. Molecular markers can effectively contribute to the formation of heterotic groups. Subsequently, parental genotypes can be selected from distinct groups with the expectation of greater heterosis effect in offspring. However, the reliability of molecular markers to confirm the relationship between genetic diversity and heterosis in offspring is neither completely clear nor universal. It depends on the type of molecular markers used, the plant species, the observed phenotypic traits, and other factors. Melchinger [6] accumulated results for numerous crops and concluded that DNA markers are well-suited for assessment of the genetic diversity, grouping of genotypes, and selecting parents to establish the base populations. Nevertheless, he [6] expected that the heterotic response between these groups cannot be simply predicted from genetic distances detected by DNA markers, but it needs to be evaluated and confirmed in real field trials. Later, the relationship between the heterosis effect and the genetic distance between parental components determined by several types of molecular markers was evaluated [7]. Among three types of molecular markers (SSR, AFLP, RAPD), the parental components for heterosis crosses were the most effectively selected by microsatellite markers (SSR) and confirmed by calculated Jaccard, Kluczyński, Nei, and Rogers coefficients, respectively [7]. Other marker types were less useful (AFLP) or declared no clear relationship between genetic distance and the heterotic effect (RAPD) [7]. The number of such studies performed in oats is very limited. However, the crossing between genotypes belonging to genetically different groups may result in high heterosis usable in breeding improvement of oat [8,9]. A genetic diversity analysis in oats themselves was performed using various molecular markers [10,11,12,13,14,15]. Relatively high polymorphism and ease of use favor SSRs located in non-coding (SSRs) as well as in coding (EST-SSRs) DNA sequences [16,17,18]. Oat breeders are also interested in the effect of heterosis obtained by the crossing of thoroughly selected parent components. An analysis of genetic dissimilarity between selected parents can be a tool to achieve this. Therefore, the aim of this study was to determine the extent of genetic diversity in a set of oats (*Avena sativa* L.) genotypes from a maintained collection of genetic resources using the microsatellite polymorphism for selection of genetically different genotypes and formation of potential heterotic groups for the oat breeding program.

## 2. Results

### 2.1. Informativeness of SSR Markers

Twenty pairs of primers for the polymorphism analysis at the microsatellite loci were tested. Only seven (AM1, AM14, AM22, AM83, AM87, AM102, AM115) revealed polymorphism within the set of 85 analyzed oat genotypes. Altogether, 35 alleles were detected, 2–10 alleles per locus, and on average 5 alleles per locus. The most polymorphic one was the locus AM1 (10 alleles) where the highest number of different microsatellite patterns (31) were found. A heterozygote status was detected at several loci.

The genetic variation at all polymorphic microsatellite loci was declared by several parameters (Table 1). The heterozygosity indices (i.e., expected heterozygosity) showed the probability that an individual genotype would be heterozygous at a given locus. Its values ranged from 0.327 to 0.5. The polymorphic information content (PIC) values showed that all seven analyzed markers (loci) had approximately the same predictive value. Although the PIC values were not high (below 0.5), together with the Heterozygosity index (H), the values suggested that primers designed for the analysis at these microsatellite loci have discriminatory competence.

The marker index (MI) is the product of the effective multiplex ratio (E) and H_avp_ and should reveal the distinguishing power of markers or techniques. It predicts the relative utility of various marker systems [19]. The MI proved to be useful in comparing the efficacy of different marker systems in some studies [20,21]. Prevost and Wilkinson [22] argued that there is little or no correlation between MI and the ability of primers to distinguish genotypes, and the practical value of MI for this purpose is limited. Only one type of marker (microsatellite) was used in our work, so the MI had no informative value in this case. Its values were low due to very low H_avp_ values.

The parameter resolving power (R) compared the diagnostic effectiveness of used primers within the analyzed set of oats. The most effective was the marker locus AM1 with the highest R value (4.1176). According to the equation of Prevost and Wilkinson [22] 0.15x + 1.78 = R (where x is the number of genotypes identified), this marker alone can identify 15 oat genotypes. Other markers had lower R values (Table 3). Even this marker is useful for discrimination of this set of oats only in combination with many others. Theoretically, at least six markers with the R value higher than 4.2 would be required for complete discrimination of all 85 evaluated oat genotypes.

The Discriminating power (D) described by Tessier et al. [23] can be a good estimator of the efficiency of individual markers or marker combinations to describe the probability that two randomly chosen genotypes have different patterns. However, the efficiency of a given marker does not depend only on the number of patterns it generates. The higher the D value (closest to 1) the lower the probability of confusion between microsatellite profiles within the analyzed oat genotypes. Both, the mean (0.8538) and individual (0.7248–0.9578) D values were high to very high in our experiment (Table 1).

Generally, the polymorphism revealed at the genomic microsatellite loci in the oat genome tends to be high. It is characterized by heterozygosity and a high number of alleles per locus, which usually results in high PIC values, diversity indices, and a high ability to differentiate genotypes [16,17,24,25]. Many of the developed EST-SSR markers have similar robustness as microsatellites (SSR) markers [26,27,28]. The effectiveness of oat genotypes distinguishing relates to the extent of the genetic diversity within the analyzed set of genotypes. This, in turn, usually depends on the geographical origin, pedigrees, and type of germplasm (cultivars, breeding lines, landraces, wild relatives). The second factor is the appropriate choice of microsatellite marker, either genomic (SSR) or expression sequence tags (EST-SSRs). The set of 85 oats used in this study contained considerable genetic variation and the parameters of all seven used polymorphic microsatellite markers had high informativeness. This allowed the subsequent analysis of genetic diversity and population structure within a set of oat genotypes.

### 2.2. Genetic Diversity

The relationships between subgroups of oat genotypes differing in seed parameters (glume color and hull/naked) faithfully demonstrated the principal component analysis (PCA). This type of statistical analysis is suitable for the discrimination of genotypes into groups and subgroups based on selected parameters. The classification of oat genotypes by PCA into subgroups according to seed parameters (white, yellow, brown, and naked) was created in this case. Four subgroups, white, yellow, brown, and naked oats, contained a different number of genotypes. A relatively high variation in analyzed microsatellite loci was revealed within each of these four subgroups. Nevertheless, it should be noted that separate subgroups overlapped and no boundaries and no separation between them were evident (Figure 1). Therefore, the relationships between the aggregating of genotypes based on variations at the microsatellite loci and the observed seed parameters could not be identified.

Although the PCA did not show the formation of separate subgroups, it revealed the differentiation ability of microsatellite markers. This analysis apparently demonstrated the suitability for differentiating genotypes of a relatively large group as a whole and individuals within four subgroups (white, yellow, brown, and nude). All genotypes within the subgroups of white, brown, and naked oats were completely differentiated from each other. Only four pairs remained unseparated within the subgroup of yellow genotypes: Hecht-Hron (PS-100), Auron-Senator, Ardo-Cyril, and Expander-Expo, respectively. Pairwise comparisons based on the PCA analysis revealed that only the brown oats were not statistically different from other subgroups of oats (Table 2)

A high differentiation competence of the microsatellite markers is better demonstrated though a cluster analysis that also indicates the possible grouping of genotypes into smaller groups/subgroups (Figure 2). It revealed the tendency for the grouping of naked and brown oats (in the middle of the dendrogram), but partially also white genotypes (in the upper half of the dendrogram). Yellow oats were scattered throughout the dendrogram.

### 2.3. Population Structure

The estimation of the population structure was performed by the STRUCTURE software using the program Structure Harvester [29]. The ∆*K* determining the best fitted value was at *K* = 2 (Figure 3). The genotypes were differentiated at *K* = 2 only into two subgroups that correspond to the two main clusters formed also by the hierarchical cluster analysis (Figure 2). There were nineteen genotypes in one subgroup (cluster at the bottom of the dendrogram, Figure 2). All the other genotypes were in the second large subgroup. However, there was no rational reason for the separation of genotypes into only two subgroups. They have different geographical origins, the period of origin, pedigrees, agronomic parameters, chemical composition of seeds, and other traits. In addition, the molecular markers used were genomic microsatellites, so it was neither reasonable to separate the genotypes into two subgroups, nor to look for associations with the observed seed parameters. Therefore, *K* = 2 was not used in the STRUCTURE analysis. The aim of the study was to create more subsets of oats that are closer to the potential heterotic groups. To achieve a more reliable and useful grouping it may be appropriate to test the *K*-values that are not the best fitted to ∆*K*. The ∆*K* helps in identifying the correct number of clusters in most situations, but it should not be used exclusively [30,31]. The *ΔK* method should be used with caution and a meaningful genetic structure should be considered when selecting an appropriate *K*-value [32]. Such an approach may allow the detection of additional substructure layer with more closely related accessions [33]. Considering these references, it was hypothesized that the formation of seven potential heterotic groups can be more applicable. The analysis conducted for *K* = 7, where the second highest peak was observed in ∆*K* plot (Figure 3), permitted a more detailed genetic grouping to seven clusters.

The alignment of individual clusters containing genotypes grouped to seven subgroups is marked in the Figure 4 (colors of clusters are the same as in the Figure 5). It reflected the microsatellite allele frequencies between subgroups (Table 3). The most distant were subgroups 3 and 7, and the most similar were 1 and 4. The average distances (expected heterozygosity) between individual genotypes within seven individual clusters indicated great variation within each of them (values are inside the Figure 4). Their values were quite comparable and showed relatively high average distances.

Seed parameters were no longer decisive for genotype classification in the presented output of the STRUCTURE analysis (Figure 5). The variation at the microsatellite loci was crucial; therefore, the STRUCTURE analysis did not present the commonly used form of clustering according to the affiliation with individual clusters. However, the grouping of genotypes according to their seed parameters (white, yellow, brown, naked) was intentionally applied. Such a presentation showed a high genetic variation within all four subgroups of oats (white, yellow, brown, and naked).

Oat genotypes with white and yellow seeds were distributed across all seven clusters. White oats had a high proportion in clusters 1 and 4 (red and yellow colors in Figure 4 and Figure 5), 24.3% and 29.7%, respectively. Yellow oats had a proportion of 28.8% in cluster 6 (turquoise) and brown oats of 43.9% proportion in cluster 4 (yellow). Naked oats were grouped mainly in cluster 7 (orange) with a proportion 42.1%. According to the frequency divergence of the analyzed microsatellite alleles, it was possible to identify the most genetically distant genotypes within individual white, yellow, brown, and naked subgroups. Among the white oat genotypes, the most distant were Edit versus Pendek, among the yellow oats Euro, Expander, Expo versus Consul, and among the naked oats genotypes Detvan versus PS-106. Within the hulled genotypes such can be Edmund, Flamingstern, and Unisignum versus Consul and Edit. For the selection of the most genetically different genotypes within the whole set of 85 oats, it would be appropriate to select genotypes from clusters 3 and 7. The classification created in this way made it possible to search the most different genotypes with identical seed parameters as well as between genotypes regardless of this parameter.

The calculated fixation indices (F*_st_*) from the STRUCTURE analysis helped to know how different subgroups can be from each other. The overall fixation index F*_st_* was 0.367, suggesting high genetic differentiation between the subgroups in the frame of the assayed germplasm and validated heterotic grouping suitable for development of advanced offspring. In addition, the F*_st_* values of seven individual subgroups were: F*_st_*_1_ = 0.181, F*_st_*_2_ = 0.412, F*_st_*_3_ = 0.498, F*_st_*_4_ = 0.434, F*_st_*_5_ = 0.214, F*_st_*_6_ = 0.490, and F*_st_*_7_ = 0.343, indicating high variation found within subgroups, especially in those included in clusters 2, 3, 4, and 6. This can suggest that the separation of genotypes according to genetic diversity at microsatellite loci can continue into even more heterotic groups. This is also supported by the ∆*K* plot (Figure 3) where another interesting peak was observed for *K* = 9.

## 3. Discussion

Melchinger and Gumber [34] in their concept of heterosis suggested approaches to identify heterotic groups. With many germplasm accessions, it is not feasible in most crops to form diallel crosses and produce sufficient F_1_ seeds for field testing in different environments. Their suggestion was to identify heterotic groups, and the first step was to group the germplasm (i.e., genotypes) according to genetic similarity. Molecular markers have been marked as extremely powerful tools for grouping of germplasm for a long time [35]. This theory was later tested in practical hybrid breeding of different plant species. Several authors have reported in different crops no or low correlations between heterosis and the genetic distance defined by molecular markers not linked to the required trait and higher heterosis effect for a yield reported in intra-group than for inter-group hybrids [36,37,38,39]. Others stated that some significant correlations between genetically different heterotic groups formed by molecular markers and yield in hybrids were found only in particular environments and across environments [40]. However, many other studies have confirmed the theory published by Melchinger [35] and Melchinger and Gumber [34]. Geng et al. [41] found that the crossing of parents from different heterotic groups, formed by SSR and SNP markers resulted in a heterotic effect in F_1_ progenies. They suggested that the genetic distance between parents determined by molecular markers can be helpful in heterosis prediction for some traits, and categorization for heterotic groups and parental selection would be beneficial in cotton hybrid breeding. Similar statements were presented in sorghum and maize [7,42,43], rye [44], barley [45], wheat [46], rice [47], and in other crops. Wang et al. [48] concluded that genomic selection predicted higher accuracy of genotypic value of genotypes for hybrid breeding but only if a higher number of field trials in more testing locations and years are used. According to them, genomic selection can be extended by marker-assisted selection using a higher number of molecular markers distributed over the genome with sufficient density. Therefore, a much simpler genetic distance analysis using different molecular markers, especially SSRs and SNPs, can be easily implemented as a simple predictor of hybrid performance in parental crosses before the implementation of such more accurate but more complicated genomic selection procedures [48,49,50].

## 4. Materials and Methods

### 4.1. Plant Material

The set of 85 oats (*Avena sativa* L.) contained registered cultivars and breeding lines from the period of 1952–2007 originating from 18 countries (including former Czechoslovakia and Soviet Union, Appendix A). The oat genotypes were with hulled or naked (hull-less) grains (*Avena sativa* var. *nuda* Koern) and differed in the color of the glume (white, yellow, brown). Seeds of all oat genotypes were obtained from the *Avena* sp. collection of genetic resources maintained in the Gene Bank of the Slovak Republic (Piešťany, Slovakia).

### 4.2. Molecular Analysis

The total DNA was extracted from young leaves using the protocol of Dellaporta et al. [51]. DNA from each oat genotype represented bulk DNA prepared from equivalent amounts of DNA from 1 to 2 g of young leaves. Twenty pairs of microsatellite-derived primers were used for analysis (Appendix A). Polymerase chain reactions were performed in 15 μL volumes containing 25 ng of DNA, 1 × PCR buffer (50 mmol/L KCl, 10 mmol/L TRIS-HCl, pH 8.3), 1.5 mmol/L MgCl_2_, 0.1 μmol/L of both primers, 0.1 μmol/L each of dNTPs, and 0.8 U of Taq-DNA polymerase. The amplification conditions included an initial denaturation 3 min at 94 °C, followed by 30 cycles of denaturation, 1 min at 94 °C, annealing 1 min at annealing temperature (Appendix A), extension 1 min at 72 °C, and the final extension 8 min at 72 °C. The equivalent volume of loading buffer (95% formamide, 10 mmol/L NaOH, 0.05% bromophenol blue) was added to each sample. Samples were denatured for 3 min at 99 °C and 5 μL of each sample was loaded into 6% polyacrylamide gel containing 7 mol/L of urea. Gels in the electrophoretic unit (SEQ 3341, Scie-Plas Ltd., Waterbeach Cambridge, United Kingdom) were run in 0.5 × TBE buffer at the constant power of 45 W for 3.5–5 h, depending on the size of amplified fragments. The microsatellite DNA was stained by the silver staining method [52]. The sizes of the microsatellite alleles were determined using 10 bp, 25 bp, 50 bp, and 100 bp DNA Ladders (Invitrogen Thermo Fisher Scientific, Waltham, MA, USA).

### 4.3. Data Analysis

The selected polymorphic indices characterizing used SSR markers and loci were calculated using the iMEC Online Marker Efficiency Calculator [21].

The principal component analysis (PCA), using the Euclidean distance measure and the cluster analysis using the neighbor joining clustering and Jaccard similarity index were performed using the Paleontological Statistics (PAST) software version 3.19 [53]. This software also used the Euclidean similarity index and 9999 permutations for the Pairwise PERMANOVA analysis using score (eigenvalues) from the first 10 significant principal components based on the Scree plot in PCA.

The population structure was evaluated through the STRUCTURE v. 2.3.4 software [54] using the default setting of the admixture model for the ancestry of individuals and correlated allele frequencies. The models were tested for *K*-values ranging from one to fifteen with ten independent runs each. The Burn-in and Markov Chain Monte Carlo iterations were set to 100,000. The number of clusters was chosen by plotting the LnP(D) values against ∆*K* values with the *K* value selected according to the Evanno test [30]. The tree in STRUCTURE was estimated using the program NEIGHBOR by Mary Kuhner and John Yamato, implementing Saitou and Nei’s neighbor joining method [55]. The plot was produced using DRAWTREE as part of his PHYLIP phylogeny package [56,57,58,59].

## 5. Conclusions

The analysis of genetic distance using genomic microsatellite markers revealed a relatively high degree of polymorphism and heterozygosity at given loci. This can be used for the almost perfect mutual differentiation of genotypes as well as for the classification of genotypes into several subgroups. Microsatellite polymorphism dominated in the way these subgroups were discriminated, but signs of grouping by seed parameters were also identifiable. High genetic differentiation between the subgroups supported a heterotic grouping. Moreover, the high heterozygosity found within subgroups suggests that microsatellite polymorphism can also be used in the formation of other heterotic groups. This approach can be implemented as a simple predictor of heterosis in parental crosses prior to extensive field testing or the development and implementation of more accurate genomic selection.

## Figures and Tables

**Figure 1 plants-10-02462-f001:**
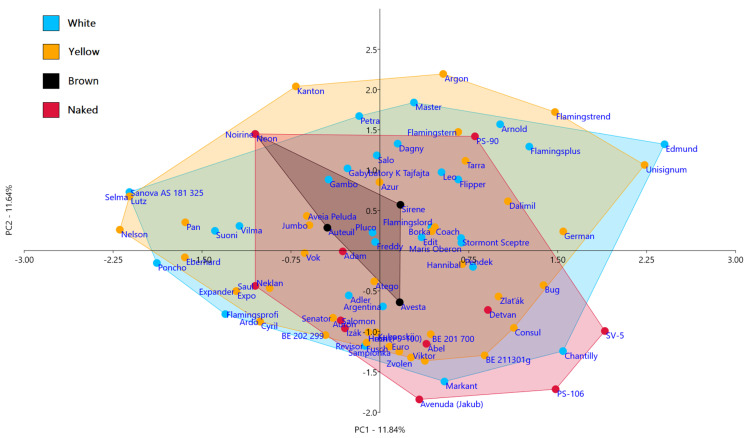
The principal component analysis (PCA) of microsatellite data and separation of oats with different seed parameters (white, yellow, brown, naked).

**Figure 2 plants-10-02462-f002:**
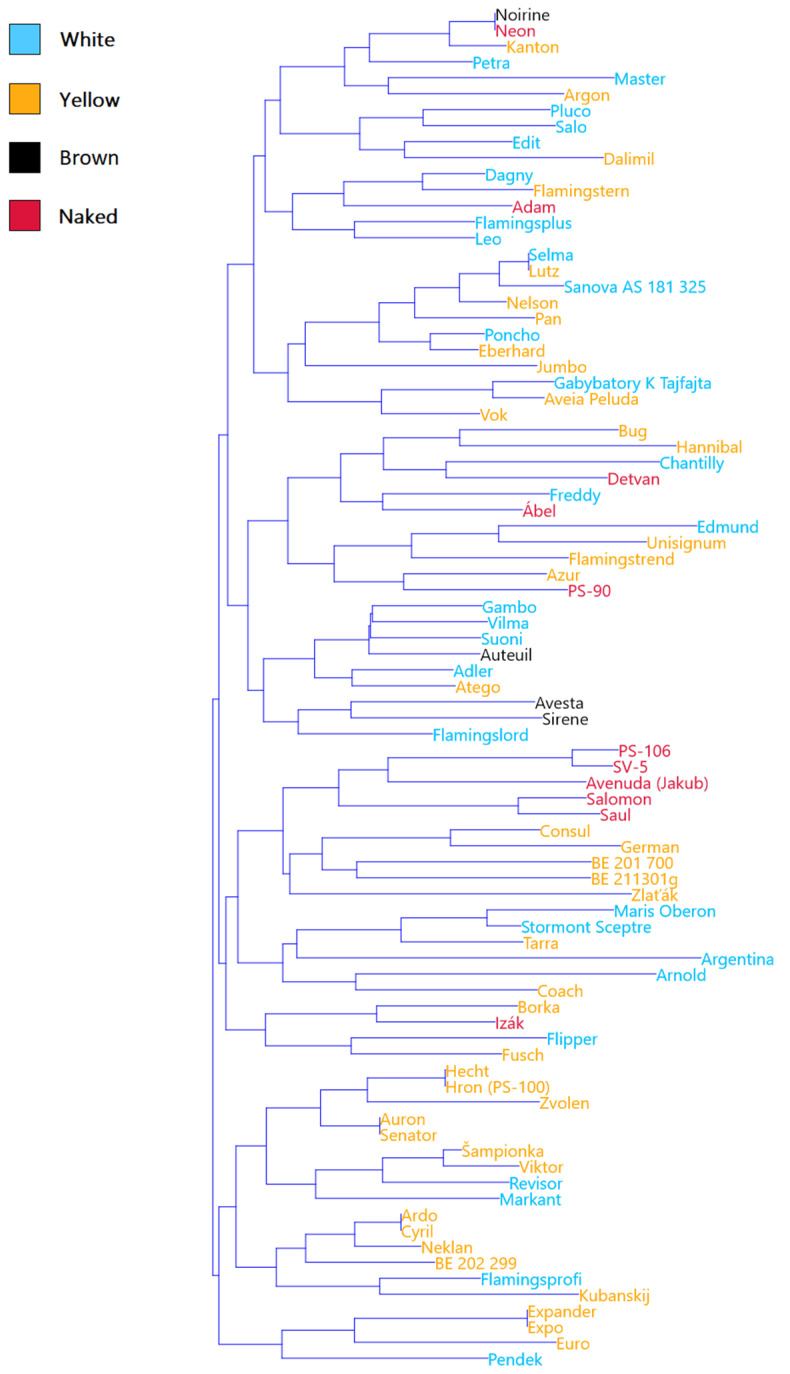
Cluster analysis of oat genotypes according to microsatellite variation.

**Figure 3 plants-10-02462-f003:**
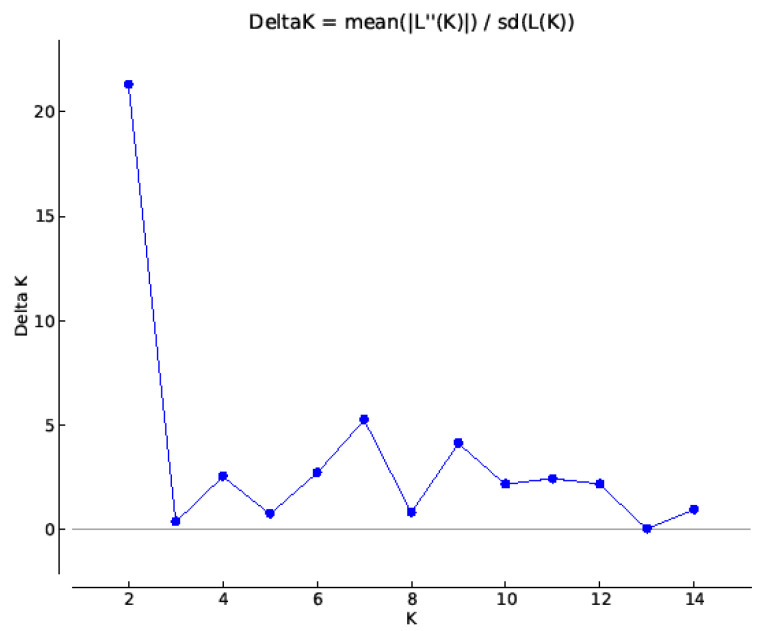
The ln(*K*) and ∆*K* values based on the dataset of used microsatellites. Hypothesized number of populations ranged from 1 to 14.

**Figure 4 plants-10-02462-f004:**
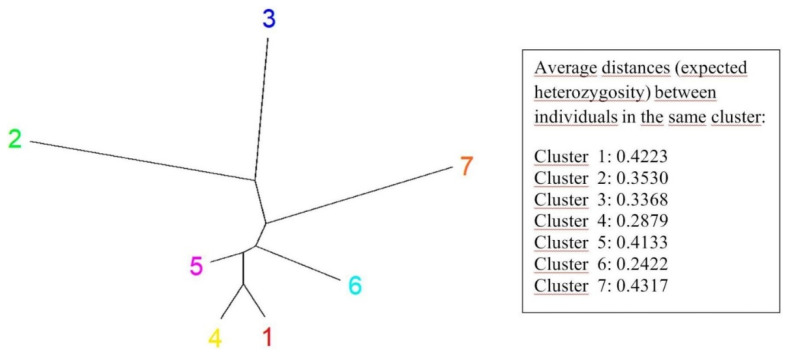
Seven clusters of grouped 85 oats according to differences in expected heterozygosity (colors of numbers correlate with colors of clusters in the Figure 5).

**Figure 5 plants-10-02462-f005:**
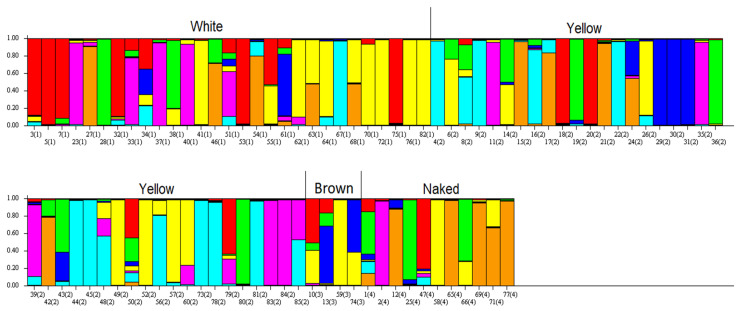
Classification of 85 oat genotypes based on microsatellite variation and seed parameters. Each bar represented individual genotype numbered in accordance with Appendix A.

**Table 1 plants-10-02462-t001:** Parameters of polymorphism in analyzed SSR loci.

Locus	n	H	PIC	E	H_avp_	MI	D	R
AM1	10	0.3270	0.3720	2.0588	0.0004	0.0008	0.9578	4.1176
AM14	8	0.3779	0.3540	2.0235	0.0006	0.0011	0.9363	2.5176
AM22	3	0.4987	0.3011	1.5765	0.0020	0.0031	0.7248	1.8353
AM83	2	0.5000	0.3004	1.0000	0.0029	0.0029	0.7515	0.1412
AM87	4	0.3750	0.3551	1.0000	0.0011	0.0011	0.9381	0.9882
AM102	4	0.3750	0.3551	1.0000	0.0011	0.0011	0.9381	0.9412
AM115	4	0.4992	0.3008	2.0824	0.0015	0.0031	0.7297	0.2118
Mean	5	0.4218	0.3341	1.5345	0.0014	0.0019	0.8538	1.5361

n—number of alleles; H—heterozygosity index; PIC—polymorphism information content; E—effective multiplex ratio; H_avp_—mean heterozygosity; MI—marker index; D—discriminating power; R—resolving power.

**Table 2 plants-10-02462-t002:** *p*-values from Pairwise PERMANOVA between all pairs of subgroups (white, yellow, brown, naked) calculated from the values of the first 10 significant principal components from PCA. Significant comparisons at *p* ≤ 0.05 are shown in bold.

	White	Yellow	Brown	Naked
White	-			
Yellow	**0.0414**	-		
Brown	0.5260	0.1204	-	
Naked	**0.0078**	**0.0014**	0.1103	-

**Table 3 plants-10-02462-t003:** Divergences in allele frequency among oat subgroups differentiated into 7 clusters.

	Cluster 1	Cluster 2	Cluster 3	Cluster 4	Cluster 5	Cluster 6	Cluster 7
Cluster 1	-						
Cluster 2	0.228	-					
Cluster 3	0.202	0.232	-				
Cluster 4	0.055	0.229	0.200	-			
Cluster 5	0.077	0.202	0.201	0.063	-		
Cluster 6	0.111	0.263	0.178	0.111	0.081	-	
Cluster 7	0.188	0.245	0.274	0.210	0.147	0.179	-

## Data Availability

Not applicable.

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
