# Peer review of "Formation of Potential Heterotic Groups of Oat Using Variation at Microsatellite Loci"

_plants, 2021, doi:10.3390/plants10112462_

Round 1
Reviewer 1 Report
General Comments:
The study entitled ‘Formation of heterotic groups of oat using variation at microsatellite loci’ by Havrlentová et al is based on exploration of genetic diversity in order to define obvious heterotic groups. To pursue this, authors have conducted a diversity analysis using multiple bioinformatics tools, but it is important to mention here that in order to define ‘heterotic’ groups, some phenotypic background should be consider (Authors have discussed this point in Introduction). Existence of genetic diversity is indeed required in plant breeding but more important is to have better performing genotypes with desirable alleles as parental groups. It would be nice to see (hopefully) in future, crosses between these defined ‘heterotic’ groups and to have an expermental validation of this study and the claimed method of developing heterotic groups.
There is a dire need to check the language of the whole manuscript by a professional and/or a native English speaker. Sentence structure sometimes do not follow what authors want to express, apparently the length of sentences also need attention, especially in the Introduction section.
Abstract: Remove the sentence ‘The Fst values of seven individual subgroups indicated high 25 variation found within each subpopulation’ from the Abstract.
Line 35: Rephrase the sentence
Line 37: Remove at the end of sentence ‘in grains’
The article as a whole requires language check from a native English speaker, specifically the Introduction part (Like 42-70).
Line 320: Replace ‘different’ with ‘differ’. There are many errors like this throughout the text. Authors are required to carefully read and make corrections wherever needed.
Author Response
Please, find our answers in attached file.

Reviewer 2 Report
This manuscript has some interesting points. I have the following comments.
Please revise the title of the manuscript.
Revise the Abstract. Please state the reasons that you undertook this research study, the objectives, the innovations and the benefits for the research community.
I personally do not find any innovations in this study. Can the authors please highlight the innovations and the importance of their research data?
There have been so many papers published on microsatellite variation and prediction of heterosis in various crops.
The authors conclude that this approach could be implemented as a simple predictor of heterosis in parental crosses prior to extensive field testing or development and implementation of more accurate genomic selection.
Please note that the authors have not demonstrated any of the above. They have just presented some data analysis but have not used it to implement a more accurate genomic selection or to predict heterosis in parental crosses. Thus, there have been no practical applications of these results.
Author Response

(The authors gave the same response as above.)

Reviewer 3 Report
The authors of manuscript (ID plants-1423690) use microsatellite markers to measure the genetic diversity of 85 oat genotypes and to group them into potential heterotic groups The obtained results provide interesting information on variation patterns and the potential of identified genotype groups to use in parental crosses for achievement of heterotic effect. In overall, the topic is of discrete interest for the readership of Plant special issue Plant Genetics, Genomics and Biotechnology. However, some aspect of the manuscript needs to be reconsidered. In particular, the results do not provide complete support to the title and part of the manuscript's conclusions. I am not sure whether crossing parents from different heterotic groups formed in this way will result in heterotic effects. As can be seen from the review of the literature on the subject, the use of genetic distances determined using molecular markers to form heterotic groups did not always yield the expected results. The relationships between the heterosis effect and the genetic distances evaluated in this way are not unambiguous. In my opinion, you can talk only about potential heterotic groups.
The manuscript is not well structured. Therefore, it is necessary to present only the results in the results section and not things from a literature review or discussion. For example, lines 89-91, “Oats (Avena sativa L.) is a hexaploid containing three 89 diploid genomes AACCDD (2n = 6x = 42 chromosomes), therefore a high number of 90 different alleles per locus and high heterozygosity were expected.“ The paragraph (lines 100-107) would be better suited to a discussion that is already very short anyway. I would say the same about the other two paragraphs: lines 116-123 and lines 140-152.
The section 2.3 also needs improvement. It presents not only results bat also discussion and methodological aspects. I would suggest structuring these things more, if there are already such separate sections.
Fig. 4 is not informative. Please construct figure based on Structure analysis with all seven clusters. Because there is talk about subpopulations, groups and clusters and it is not clear does these terms mean the same or different things. I would suggest to use term “group” or “cluster”. If subpopulations, groups and clusters are different entities, this should be indicated in the methodological section. There is no real basis for talking about subpopulations as well as about the population here.
Aso, I did not understand the meaning of the last sentence (lines 278-280) of section 2.3. After all, before that, there was also a talk about seven clusters.
The last sentences of the discussion (lines 307-314) are, in my view, very similar in their sense, so one of them should be deleted or rewritten.
I also have comments on the methodological part. It is not clear from the description from which collection or gene bank they are received. Is it collection of National Agricultural and Food Center?
I did not find information in section 4.2 on how the size of DNA fragment was determined; how the initial matrix was formed for further calculations. Whether SSR markers were considered dominant or codominant in it. Oats are polyploids. There is also no information anywhere on the repeatability of the analyzes, how it was assessed. It is not specified what the specific diversity parameters were evaluated and how they were calculated. What are their abbreviations. They appear in Table 3, but there is no detailed explanation in the methodological part. It is also unclear how Fst1-7 values were obtained and what they indicate.
Author Response

(The authors gave the same response as above.)

Reviewer 4 Report
The article is very interesting and the results obtained necessary to use in breeding practice. The use of genotyping with SSR markers is a very informative method for identifying phenotypic diversity in oats. It presents data on the genetic diversity of a large set of cultivated oat varieties with different colors of the glume and nakedness.
At the same time, I would like to make a number of comments.
- The first remark about who or what was the donor of the studied material (genebank, institute, collection).
- In the study, signs were taken (color of the glum and nakedness), which indicate the intraspecific diversity of this species. It would like these forms to be correctly identified from a botanical point of view.
- In addition, according oats have not three types of colors, but four − these are white, yellow, grey (black) and brown. Despite the fact that each of them has awns and awnless forms, which are not mentioned here.
- Inspection of 4 varieties with "black" color showed that two of them are brown − Avesta and Sirene.
- Also, out of 85 varieties of the studied set, only four varieties had a "black" color, which is not statistically comparable with the results of other groups for the color of a much larger volume.
Author Response

(The authors gave the same response as above.)

Round 2
Reviewer 2 Report
The authors have replied to my comments and improved their manuscript.
Author Response
Answers are attached.

Reviewer 3 Report
Thank you for the answers. In my opinion, some problems with reproducibility exist for all types of molecular and biological assays, and SSRs are not exception. Please read, Bonin et al., 2004. How to track and assess genotyping errors….” Mol. Ecology, 13, 3261-3276. “Genotyping errors can be generated at every step of the genotyping process (sampling, DNA extraction, molecular analysis, scoring, data analysis) and by a variety of factors (chance, human causes, technical artefacts) “. Besides, I also would recommend to include information about the size of identified alleles in to Table S2 and put the appropriate link in the results section (line 99 may be?), because this information is important for your study.
Author Response
Answers are attached.
